# MXene–AuNP-Based Electrochemical Aptasensor for Ultra-Sensitive Detection of Chloramphenicol in Honey

**DOI:** 10.3390/molecules27061871

**Published:** 2022-03-14

**Authors:** Jing Yang, Wei Zhong, Qi Yu, Jin Zou, Yansha Gao, Shuwu Liu, Songbai Zhang, Xiaoqiang Wang, Limin Lu

**Affiliations:** 1Key Laboratory of Crop Physiology, Ecology and Genetic Breeding, Ministry of Education, Key Laboratory of Chemical Utilization of Plant Resources of Nanchang, College of Chemistry and Materials, Jiangxi Agricultural University, Nanchang 330045, China; yangjing202195@163.com (J.Y.); zhongwei2020hai@163.com (W.Z.); qiyujxau@163.com (Q.Y.); 15797710413@163.com (J.Z.); suwu762846210@126.com (S.L.); xqwang79@sina.com (X.W.); 2Hunan Provincial Key Laboratory of Water Treatment Functional Materials, Hunan Province Engineering Research Center of Electroplating Wastewater Reuse Technology, College of Chemistry and Materials Engineering, Hunan University of Arts and Science, Changde 415000, China

**Keywords:** MXene, gold nanoparticle, chloramphenicol, electrochemical, aptasensor

## Abstract

A simple and label-free electrochemical aptasensor was developed for ultra-sensitive determination of chloramphenicol (CAP) based on a 2D transition of metal carbides (MXene) loaded with gold nanoparticles (AuNPs). The embedded AuNPs not only inhibit the aggregation of MXene sheets, but also improve the quantity of active sites and electronic conductivity. The aptamers (Apts) were able to immobilize on the MXene–AuNP modified electrode surface through Au–S interaction. Upon specifically binding with CAP with high affinity, the CAP–Apt complexes produced low conductivity on the aptasensor surface, leading to a decreased electrochemical signal. The resulting current change was quantitatively correlated with CAP concentration. Under optimized experimental conditions, the constructed aptasensor exhibited a good linear relationship within a wide range of 0.0001–10 nM and with a low detection limit of 0.03 pM for CAP. Moreover, the developed aptasensor has been applied to the determination of CAP concentration in honey samples with satisfactory results.

## 1. Introduction

As the first synthetic and mass-produced antibiotic, chloramphenicol (CAP) was first discovered in *Streptomyces Venezuelanus* in 1947 [1], which is extensively used to treat several infectious diseases in animals due to its exceptional antibacterial and great toleration [2]. However, abuse leads to CAP drug residue in foods of animal origin such as honey, which has significant potential side effects in humans, e.g., aplastic anemia, leukemia, gray-baby syndrome and so on [3,4]. For these reasons, several countries have forbidden the use of CAP in humans and food manufacture [5]. At the same time, the European Union (EU) has defined the uppermost CAP concentration as 0.3 μg·kg^−1^ in products of animal origin [6]. Nevertheless, CAP can still be detected in some animal source foods such as honey, owing to its low cost and easy availability. Thus, the development of sensitive analytical methods for detecting CAP drug residue in honey is urgent.

So far, several techniques for detecting CAP drug residue have been developed, for example, photoinduced chemiluminescence, high-performance liquid chromatography (HPLC) [7], liquid chromatography (LC) [8], liquid chromatography–mass spectrometry (LC-MS) [9], and gas chromatography–mass spectrometry (GC-MS) [10] etc. Although these methods are reliable, they are accompanied by the disadvantages of cumbersome instrumentation, complicated operation, and high cost. Compared with these traditional approaches, the electrochemical analysis method is widely used because of its merits of simplicity, fast response time, low cost, real-time monitoring and miniaturization [11,12].

When constructing an electrochemical sensor for detecting CAP drug residue, researchers should consider the target molecule recognition element to selectively recognize the target molecule on the one hand, and the signal amplification methods to realize the sensitive analysis of the target molecule on the other. For molecular recognition, antibodies are the most used recognition element in constructing electrochemical biosensors for drug residue detection [13]. In recent years, nucleic acid aptamers have been widely used as target molecular recognition elements in biosensor construction. Aptamers (Apts) are short synthetic single-stranded DNA or RNA oligonucleotides that optionally bind to a variety of targets including proteins, metal ions, and small molecule toxins [14], etc. Presently, a vast number of aptamers have been chosen successfully and applied to build electrochemical aptasensors for CAP [15,16]. Since CAP drug residue is usually low, signal amplification method is usually needed to improve the detection sensitivity of target molecules. In the construction of electrochemical biosensors, the use of various functional materials is a common method for signal amplification since appropriate nanomaterial-modified sensing interfaces can effectively improve detection performance [17]. Till now, several nanomaterial-based electrochemical aptasensors, such as polyethyleneimine-reduced graphene oxide/gold nanocubes [18], Ag nanoparticle-sensitized bismuth oxyiodide [19], and gold nanoparticle-functionalized 3D flower-like MoS_2_/TiO_2_ [20], have been reported for CAP detection. Nevertheless, the sensitivity of these sensors still needs to be improved. Hence, it is imperative to exploit new electrochemical sensing substrates.

Recently, two-dimensional (2D) MXenes have received great interest for their large group of carbides, nitrides, and carbonitrides, and for their unique properties such as high electrical conductivity, large specific surface areas and superior electrochemical properties [21]. As the most widely studied type of MXene, Ti_3_C_2_T_x_ possesses excellent hydrophilicity, high specific surface area, and good electrical conductivity, which makes it a promising candidate in electrochemical sensing applications [22]. However, due to the stacking of van der Waals forces, MXene nanosheets lose several exposed active sites, which limits their electrochemical performance [23]. Recent research has indicated that the introduction of interlayer spacers, metal nanoparticles (NPs), between the sheets can effectively avoid MXene aggregation [24]. A common method is to use the strong reducibility of MXene to reduce metal salts to form the composite of MXene and metal nanoparticles. For instance, our group put forward the use of a composite of MXene@Ag nanoclusters (AgNCs) and amino-functionalized multi-walled carbon nanotubes as a sensing platform for electrochemical detection of carbendazim, where well-dispersed AgNCs were allowed to grow in situ on the surface of MXene with self-reduction approach [25]. Furthermore, studies have demonstrated that gold nanoparticles (AuNPs) could furnish more active sites to immobilize aptamers due to its characteristics, including high specific surface area, good biocompatibility and catalytic properties [26]. According to this, as an outstanding substrate and an excellent immobilization matrix, MXene modified by AuNPs would be quite suitable as an electrochemical aptasensor.

Based on the above, we developed a simple and label-free electrochemical aptasensor based on AuNP-modified MXene sheets, which can implement efficient and sensitive detection of CAP. AuNPs grew well on the surface and between the layers of MXene through a self-reduction process without reducing agent or stabilizer. As a promising sensing platform, the composite of MXene–AuNPs possesses a high specific surface area, abundance of exposed active sites, and outstanding electronic conductivity. The Apt with 5′-end modified sulfhydryl was incubated on the surface of a MXene–AuNP-modified electrode by Au–S bond. When CAP was added, CAP–Apt complexes produced with poor conductivity on the surface of sensor increased the impediment of electron transfer, causing a decrease in electrochemical signal. The designed electrochemical aptasensor for CAP detection presents a wide linear range and a low detection limit. In addition, the aptasensor displays considerable potential in practical sample analysis with excellent recovery.

## 2. Results and Discussion

### 2.1. Sensing Mechanism of the CAP Electrochemical Aptasensor

The fabrication process of the aptasensor and the sensing mechanism of target molecules are represented in Figure 1. Making use of the strong reducibility of MXene, AuNPs were synthesized on the MXene sheets to obtain a MXene–AuNP composite using an in situ synthesis method. By simply dropping the resulting MXene–AuNP composite on the GCE electrode surface, the electrode could be easily modified with MXene–AuNPs, generating an efficient electrode surface with larger surface area and more active sites. The CAP aptamer modified by thiol was fixed on the surface of the MXene–AuNP electrode via Au–S interaction, which was used as an identification element for target molecules. The non-specific sites of the electrode surface were blocked by self-assembly of MCH. In the existence of the CAP target molecule, the aptamer switched its structure to specifically bind with CAP, which leads to significant blockage of electron transfer. Thus, the detection of CAP can be achieved easily by monitoring the electrochemical current decrease of the developed aptasensor, demonstrating that it is a new and simple CAP sensor platform.

### 2.2. Material Characterization

The surface morphologies of MXene and MXene–AuNPs were studied by SEM. As shown in Figure 1A, a representative accordion-like morphology was observed, and this particular 2D multilayer structure of MXene can provide a great specific surface area. After mixing HAuCl_4_ with MXene, it can be observed that AuNPs were evenly distributed on the surface and interlayer of MXene with an average size of 50 nm via a self-reduction process (Figure 1B). Here, embedding with AuNPs can markedly improve the conductivity and enhanced electrochemical properties of MXene, further developing its application as an electrochemical aptasensor for immobilizing the CAP aptamer. The XRD spectra of MXene and MXene–AuNPs are shown in Figure 1C. There are four characteristic main peaks in the XRD pattern of MXene–AuNPs, which belong to the (111), (200), (220) and (331) lattice planes of face-centered cubic Au single crystals [27].

### 2.3. Electrochemical Characterization of the Aptasensor

Cyclic voltammogram (CV) and electrochemical impedance spectroscope (EIS) techniques were used to characterize each step of the CAP aptasensor fabrication procedure. The CV curves for different steps of modified electrodes were recorded in [Fe (CN)_6_]^3−/4−^ redox probe solution. As shown in Figure 2A, the bare GCE electrode (curve a) exhibited a well-defined pair of reversible redox peaks. Compared with bare GCE, the reversible redox peak current increased markedly, and the peak potential separation (ΔEp) decreased after immobilization of MXene–AuNPs (curve b). This can be attributed to the composite of MXene–AuNPs, which can effectively increase the conductivity of the electrode and facilitate electron transfer. After Apt was incubated on the MXene–AuNP/GCE surface, the redox peak current dropped dramatically, indicating that the negatively charged phosphate backbone of the aptamer exerted strong electrostatic repulsion forces to [Fe(CN)_6_]^3−^/^4−^ (curve c) [28]. After treatment with MCH, a further decrease was observed in peak current due to the insulating property of MCH (curve d). Finally, when the CAP target was added, CAP specifically bonded with Apt and led to a further decrease in CV response (curve e). This could be attributed to the fact that the specific binding of CAP and Apt causes a configuration change of Apt to impede the charge transfer process. As a result, all observations of CV showed that the Apt/MXene–AuNP/GCE was successfully constructed.

During the construction of the CAP aptasensor, the EIS characterization of various modified electrodes is shown in Figure 2B, which exhibits typical Nyquist plots of bare GCE (curve a), MXene–AuNP/GCE (curve b), Apt/MXene-AuNP/GCE (curve c), MCH/Apt/MXene-AuNP/GCE (curve d) and CAP/MCH/Apt/MXene-AuNP/GCE (curve e) in [Fe (CN)_6_]^3−/4−^ mixture solution (1:1) containing 0.1 M KCl. In the EIS, the higher-frequency semicircle corresponds to the electron-transfer-limited process, and its diameter is equal to the electron-transfer resistance [18]. A semicircle with smaller resistance value was observed at MXene–AuNP/GCE (curve b) than the bare GCE (curve a). This is because of the good electrical conductivity of the prepared MXene–AuNP composites, which increases the electroactive sites of the electrode and boosts electron transfer on the electrode surface. By incubating the Apt to MXene–AuNP/GCE, the resistance of Apt/MXene-AuNPs/GCE increases because the negative charge on the ssDNA phosphate backbone that could impede the electron transfer of [Fe(CN)_6_]^3−/4−^ (curve c). After the introduction of MCH, the resistance increases gradually due to its non-electroactive property (curve d). When the CAP target was combined with the MCH/Apt/MXene-AuNPs/GCE surface, the particular binding of CAP with Apt formed complexes in which the electron-transfer block increased owing to steric hindrance. The results of EIS were consistent with the CV results, which certified that the CAP electrochemical aptasensor had been successfully constructed.

### 2.4. Optimization of Experimental Conditions

To enhance the detection capability of the developed aptasensor, experimental conditions including the volume of MXene–AuNPs, the concentration of Apt, the incubation time of Apt and the interaction time between CAP and Apt were optimized. As a key index for evaluation of aptasensor performance, the inhibition ratio Q (ΔI/I_0_) of a certain concentration of CAP can be calculated as follows:ΔI/I_0_ (%) = [(I_0_ − I_1_)/I_0_] × 100
where I_0_ and I_1_ represent the current of MCH/Apt/MXene-AuNP/GCE and CAP/MCH/Apt/MXene-AuNP/GCE. As can be seen in Figure 3A, the inhibition ratios of modified electrodes with different volumes of MXene–AuNPs were first studied. It can be seen that the inhibition ratio increased as the volume of materials changed from 3 to 5 μL due to the increased active sites on the surface of MXene–AuNP/GCE. However, the inhibition ratio decreased progressively as the modified volume exceeded 5 μL, which might be related to excessive MXene–AuNPs and will cause accumulation and blockage of electron transfer. Therefore, 5 μL was adopted for subsequent experiments.

The Apt density on the electrode surface directly influenced the current response signal of the aptasensor. Therefore, the concentration and incubation time of the Apt were investigated, as shown in Figure 3B,C. As the concentration of Apt increased, the inhibition ratio reached a maximum at 1.0 μM and then decreased gradually. This can be attributed Apt continuously binding with MXene–AuNPs after saturation, leading to the steric hindrance effect derived from excessive binding. The incubation time of Apt is another significant factor that affects the performance of the aptasensor. The results of experiments with a sequence of Apt incubation times (2, 4, 6, 8, 10 h) were compared in Figure 3C. It can be noted that the maximum inhibition ratio was acquired when the incubation time was 4 h. Although the incubation time of Apt is too long, overstocked Apt decorating the electrode surface might cause large steric hindrance, leading to a decrease of the inhibition ratio. Thus, 4 h was selected as the best Apt incubation time.

Furthermore, the reaction time between Apt and CAP target molecules also influenced the performance of the aptasensor directly. As shown in Figure 3D, the inhibition ratio increased rapidly with the increase of reaction time until 120 min and then stayed steady. The results indicated that the formation of the Apt–CAP complex finished after 120 min. Thus, 120 min was adopted as the optimized response time for Apt and CAP.

### 2.5. Analytical Performance of the Aptasensor

Under optimal conditions, the detection performance of the Apt/MXene-AuNP/GCE aptasensor for CAP was investigated by DPV. Figure 4A shows the DPV curves of the Apt/MXene-AuNP/GCE to CAP with different concentrations (0–10 nM). The oxidation peak currents decreased with increasing CAP concentration due to fact that the formation of Apt–CAP complex, which acts as a barrier film preventing the electron exchange on the surface of the electrode. A linear relationship between ΔI value and the logarithmic of CAP concentration was obtained in a wide range of 0.0001–10 nM, as shown in Figure 4B. The linear equation was ΔI (μA) =17.958 + 11.201 lg C (pM) (R^2^ = 0.996) and the limit of detection (LOD) was calculated as 0.03 pM (based on the signal-to-noise ratio [S/N] = 3). Compared with some other CAP aptasensors reported in the literature (Table 1), the developed aptasensor based on MXene–AuNP/GCE for CAP detection exhibits high sensitivity. 

### 2.6. Selectivity, Reproductivity, and Stability Investigation

The selectivity of the constructed aptasensor for CAP detection in similar species was also investigated. Four antibiotics—magnolol (MAG), midecamycin (MID), isoniazid (INH) and diethylstilbestrol (DES)—were selected to evaluate the selectivity of the proposed aptasensor. As shown in Figure 5A,B, the CAP target with a low concentration of 1 nM generated a high inhibition ratio of about 56%, while other antibiotics with much higher concentration of 50 nM only induced very low inhibition ratio, indicating that the designed aptasensor showed promising selectivity for CAP. 

Reproducibility and long-term stability are also two important indicators of the inductor. The reproducibility of the biosensor was evaluated using five different GCE to detect the 1 nM CAP with identical manufacturing processes (three measurements for each electrode). The relative standard deviation is less than 3.59%, indicating that the designed sensing strategy has good reproducibility (see Figure 5C). For long-term stability investigation, a batch of electrodes was treated with the same fabrication procedure to obtain the sensing interface and stored in the refrigerator. The 1 nM CAP sample was measured day by day and the results are shown in Figure 5D. After 15 days, the signal value remained at 95% of the initial value, demonstrating the excellent stability of developed aptasensor.

### 2.7. Analysis of Spiked Samples

To further investigate the performance of the proposed aptasensor for practical applications, spiking and recovery studies were used to determine the amount of CAP in honey samples. The honey samples were purchased from the local supermarket and processed as described in Section 2.3. The detection results are summarized in Table 2. The mean recoveries were calculated in a range from 95.76% to 106% (*n* = 5) with RSDs below 4.2%. Moreover, the results are in good agreement with the high-performance liquid chromatography (HPLC) method, which indicates the satisfactory potential of the designed aptasensor for the detection of CAP in practical samples.

## 3. Materials and Methods

### 3.1. Reagents and Materials

Ti_3_C_2_T_x_ MXene powder was purchased from Nanjing XF Nano Co., Ltd., (Nanjing, China). HAuCl_4_, 6-Mercapto-1-hexanol (MCH), Tris-HCl, EDTA, K_4_[Fe(CN)_6_], K_3_[Fe(CN)_6_] and KCl were obtained from Aladdin Biotechnology Co., Ltd. (Shanghai, China). Chloramphenicol (CAP), magnolol (MAG), midecamycin (MID), isoniazid (INH) and diethylstilbestrol (DES) acquired by Yuanye Bio-Technology Co., Ltd. (Shanghai, China). All other reagents and solvents used were of analytical grade. All aqueous solutions were made with Millipore-Q water. Phosphate buffered saline (PBS, 10 mM, pH 7.4) was prepared with KCl, Na_2_HPO_4_ and NaH_2_PO_4_ solutions to solubilize the [Fe(CN)_6_]^3−/4−^ redox couple for the electrochemical determinations. MCH was dissolved in Tris-HCl buffer (30 mM Tris-HCl, 4 mM MgCl_2_, 300 mM KCl, pH = 7.8) to attain 1 mM stock solutions. 

The sequence of CAP-binding aptamer (Apt) was obtained from Sangon Biotechnology Co., Ltd. (Shanghai, China), and its sequence is as follows: 5′-C6/SH-ACT TCA GTG AGT TGT CCC ACG GTCGGC GAG TCG GTG GTA G-3′, which was dissolved in 1 × TE buffer (pH = 7.6, 10 mM Tris-HCl, 1 mM EDTA) to attain stock solutions with a certain concentration [18]. 

### 3.2. Instruments

Scanning electron microscopy (SEM, Hitachi S3400 N, Tokyo, Japan) and X-ray diffraction (XRD, Bruker D-8 X-ray diffractometer, Karlsruhe, Germany) were used for characterization. Electrochemical measurements were adapted from an electrochemical workstation (CHI-750E, Shanghai, China) with a standard three-electrode system consisting of working electrode (GCE, *Φ* = 3 mm), saturated calomel electrode (SCE) and platinum wire electrode.

### 3.3. Preparation of MXene–AuNP Composite

MXene–AuNPs were synthesized based on previous literature with minor modification [27]. The brief steps were as follows: 6 mg of MXene was ultrasonically dissolved in 3 mL ultrapure water for 30 min to obtain a uniform solution. Subsequently, 500 μL of HAuCl_4_ aqueous solution (30 μM) was added slowly by mixing gently. After 10 min of reaction, the MXene–AuNP suspension was centrifuged at 8000 rpm for 10 min and the gathered precipitate was re-dispersed in ethanol for further testing.

### 3.4. Preparation of Aptasensor

GCE surface was polished with 0.3 μm and 0.05 μm alumina powder sequentially, then washed with deionized water and sonicated in ethanol/water for 5 min. Next, 5 μL of MXene–AuNP suspension (2 mg mL^−1^) was immobilized on the surface of bare GCE and under infrared-lamp drying to obtain MXene–AuNP/GCE. A total of 10 μL of Apt solution (1 μM) was cast on the resulting electrode and incubated for 4 h at 37 °C, then washed with ultrapure water to clear away unbound aptamer. Subsequently, 5 μL of MCH solution (1 mM) was dropped and incubated for 20 min at 37 °C to block the non-specific binding sites. After washing with ultrapure water to clear away the unbound MCH, the aptasensor was finally obtained and stored at 4 °C until use.

### 3.5. Honey Sample Preparation

Honey samples purchased from the local mart were pre-treated following the previously reported method [34]. Simply, 1.0 g of honey was placed in a plastic centrifuge tube and mixed with 3 mL of PBS. Then, an appropriate amount of CAP was added to the honey sample without CAP and 4 mL of ethyl acetate was dropped into the honey sample. After incubation for 15 min, the resulting samples were centrifuged for 10 min at 5000 rpm and transferred the supernatant to another clean centrifuge tube. The solutions containing CAP were stored in the refrigerator for subsequent use.

### 3.6. Electrochemical Measurements

Electrochemical impedance spectroscopy (EIS) measurements were performed in 5.0 mM K_3_[Fe(CN)_6_]/K_4_[Fe(CN)_6_] solution containing 0.10 M KCl with a frequency range of 0.01 Hz to 100 kHz and an amplitude of 5.0 mV. Differential pulse voltammetry (DPV) measurements were recorded in 5.0 mM K_3_[Fe(CN)_6_]/K_4_[Fe(CN)_6_] solution containing 0.10 M KCl in the potential range of −0.2–0.6 V. Cyclic voltammetry (CV) measurements (potential range of −0.2–0.6 V, scan rate of 100 mV s^−1^) were obtained to test the performance of the prepared electrodes.

## 4. Conclusions

In summary, a simple and sensitive aptasensor based on MXene–AuNP composite material for the electrochemical detection of CAP was developed. The MXene–AuNP composite exhibited large specific surface area, abundant exposed active sites, prominent biocompatibility and outstanding electronic conductivity, which facilitate the immobilization of the Apt on the electrode surface. The interaction of the Apt and CAP was investigated by CV, EIS and DPV electrochemical techniques. In the presence of CAP, the Apt–CAP complex was formed on the surface of MXene–AuNP/GCE, which created a hindrance or restriction to the exchange of electrons. The presented aptasensor exhibits high sensitivity and selectivity, good reproducibility and stability, with a low LOD of 0.03 pM and a wide linear range of 0.0001–10 nM. More importantly, the aptasensor further shows its detection ability in honey samples with satisfactory results, suggesting great potential for the in situ analysis of antibiotic residues.

## Data Availability

The data presented in this study are available in article.

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
