# Peer review of "MXene–AuNP-Based Electrochemical Aptasensor for Ultra-Sensitive Detection of Chloramphenicol in Honey"

_molecules, 2022, doi:10.3390/molecules27061871_

Round 1

Reviewer 1 Report

This manuscript describes an aptasensor based on 2-dimensional MXene-AuNPs films for the detection of the antibiotic chloramphenicol in honey samples. The work is in general well described, presented and discussed and I recommend acceptance after addressing the following points.

Comments:

  • The sensing surface, based on AuNPs and aptamers, and the sensing mechanism, based on the inhibition of ferricyanide signal, are not new. Please comment on the originality and novelty of the work in the Introduction.
  • Is the electrode surface completely covered? A SEM image of the electrode surface, showing the coverage of MXene/AuNPs must be presented.
  • Please explain how the 5 mM concentration of ferricyanide was selected.
  • Section 2.7 does not correspond to real sample analysis but to analysis of spiked samples. Please correct.
  • What is the stability of the biosensors? Can they be reused after regeneration?

Author Response

Comments of reviewer #1 as well as our revisions and replies are as follows:

  1. The sensing surface, based on AuNPs and aptamers, and the sensing mechanism, based on the inhibition of ferricyanide signal, are not new. Please comment on the originality and novelty of the work in the Introduction.

Response: Thanks for the reviewer’s kind suggestion. We agree that sensing mechanism based on aptamers immobilized on AuNPs integrated with the inhibition of ferricyanide signal has been reported. Till now, several nanomaterials-based electrochemical aptasensors, such as polyethyleneimine reduced graphene oxide/gold nanocubes [18], Ag nanoparticles sensitized bismuth oxyiodide [19], and gold nanoparticle-functionalized 3D flower-like MoS2/TiO2 [20], have been reported for CAP detection. Nevertheless, the sensitivity of these sensors still needs to be improved. Hence, it is imperative to exploit new electrochemical sensing substrates.

In this work, a new composite material of MXene-AuNPs was used AuNPs as the sensing substrate. MXene, the 2D transition metal carbides, possesses excellent hydrophilicity, high specific surface area and good electrical conductivity, which make them promising candidates in electrochemical sensing applications. This work developed a simple and efficient aptamer immobilization method and electrochemical sensing interface based on MXene-AuNPs composite material for the first-time using chloramphenicol (CAP) as the model target. The embedded AuNPs not only inhibit the aggregation of MXene sheets, but also improve the quantity of active sites and electronic conductivity. The aptamers (Apts) were able to immobilize on the MXene-AuNPs modified electrode surface through Au–S interaction, and the constructed electrochemical aptasensor showed excellent specificity, good reproducibility, and acceptable stability. Therefore, this MXene-AuNPs based sensing strategy might provide an effective candidate for the detection of trace level of chloramphenicol. We have added the originality and novelty of this work as shown in red in the Introduction section of the revised manuscript [p.2, line 89-96].

  1. Is the electrode surface completely covered? A SEM image of the electrode surface, showing the coverage of MXene/AuNPs must be presented.

Response: For aptasensor preparation, 5 μL of 2 mg mL-1 MXene-AuNPs suspension was dipped onto the surface of bare GCE, which theoretically completely covered the electrode surface. To observe the coverage of MXene-AuNPs composite on the electrode surface, we have performed the experiment and the SEM images show that the MXene-AuNPs composite are uniformly coated on the electrode surface.

Supplementary Figure (not contained in the manuscript). SEM image of MXene-AuNPs composite on the electrode surface using conductive glass with ITO film as the simulated substrate.

  1. Please explain how the 5 mM concentration of ferricyanide was selected.

Response: Thanks for the reviewer’s kind suggestion. The 5 mM concentration of ferricyanide was selected according to the previous reported references. In fact, the concentration of 5 mM is a commonly used parameter when using ferricyanide as the electrolyte in electrochemical measurement. Some references as listed as following.

[1] Wang, L.; Jin, H.; Wei, M.; Ren, W.; Zhang, Y.; Jiang, L.; Wei, T.; and He, B. A DNAzyme-assisted triple-amplified electrochemical aptasensor for ultra-sensitive detection of T-2 toxin. Sens. Actuators B Chem. 2021, 328, 129063. 10.1016/j.snb.2020.129063.

[2] Zeng, G.; Zhang, C.; Huang, D., Lai; C., Tang; L., Zhou; Y., Xu; P., Wang; H., Qin; L., Cheng, M. Practical and regenerable electrochemical aptasensor based on nanoporous gold and thymine-Hg2+ -thymine base pairs for Hg2+ detection. Biosens. Bioelectron. 2017, 90, 542-548. 10.1016/j.bios.2016.10.018.

  1. Section 2.7 does not correspond to real sample analysis but to analysis of spiked samples. Please correct.

Response: Thanks for the reviewer’s kind suggestion. Inappropriate expressions have been corrected in the Section 2.7 as shown in red in the revised manuscript [p 11, line 356].

  1. What is the stability of the biosensors? Can they be reused after regeneration?

Response: Thanks for the reviewer’s kind suggestion. Since the fabrication procedure of the sensing interface is easy, the electrode was polished after each measurement. The reproducibility of the biosensor was evaluated by using five different GCEs to detect 1 nM CAP with the identical manufacturing processes (three measurements for each electrode). The relative standard deviation is less than 3.59%, indicating the designed sensing strategy has good reproducibility (seen in Fig. 5C).

For long-term stability investigation, a batch of electrodes were treated with the same fabrication procedure to obtain the sensing interface and stored in the refrigerator. The 1 nM CAP sample was measured day by day and the results were shown in Fig. 5D. After 15 days, the signal value remained 95% of the initial value, demonstrating excellent stability of developed aptasensor.

Fig. 5 (A) The inhibition ratio of the aptasensor in presence of 1 nM CAP or 50 nM interferences (MAG, MID, INH and DES), (B) DPV response of the aptasensor in presence of 1 nM CAP or 50 nM interferences (MAG, MID, INH and DES), (C) Reproducibility of the aptasensor, (D) Stability of the aptasensor.

Reviewer 2 Report

In the present manuscript, the authors presented the development of an electrochemical aptasensor based on 2D based nanocomposite material consisting by MXene and AuNP for sensitive detection of chloramphenicol. 

Even if the presented results revealed a very low detection limit and a reduced linear range, apparently beneficial for the detection of trace level of chloramphenicol in real samples, these are some issues to be clarify, and I consider that the present results must be completed with some other details.

Chapter 1 - Introduction: Lines 59 - 60 There should be added more references regarding the electrochemical aptasensors developed recently in literature, not only for food samples,. Few more examples should be added based on the novel nanomaterials.

Chapter 2.3. - Electrochemical characterization of the aptasensor - Here and neither in Chapter 3 Material and methods aren't presented the specific parameters used for EIS (frequency range, amplitude, etc) and for DPV (working applied potential, etc). 

Chapter 2.4 - Optimization of experimental conditions - The specific procedures for preparation of this aptasensor require long time, more than 6 hour, which is a big disadvantage, and since there are not presented studies regarding the stability and reproducibility of this developed aptasensor, raises doubts about the accuracy of the analytical parameters obtained. 

Chapter 2.5. - Should be completed with stability and reproducibility of the developed aptasensor, the storage life of it, etc.

Chapter 2.6. Selectivity studies - Figure 5 should be completed also with DPV (currents plots) for specific species using the reported concentrations.

Chapter 2.7.  I recommend validation test with another technique for the real samples of honey and spiked honey (pM concentrations).

Chapter 3 - Should be completed with correct information about electrochemical system (reference and counter electrodes used), specific parameters used for EIS, DPV and CV (potential range, scan rate, etc).

Chapter 4 - Should be completed in accordance with information about stability, reproducibility, response time and life time of the developed aptasensors, and which are the advantages of presented aptasensor compared with reported ones, in terms not only of limit of detection and linear range. 

Author Response

Comments of reviewer #2 as well as our revisions and replies are as follows:

  1. Chapter 1 - Introduction: Lines 59 - 60 There should be added more references regarding the electrochemical aptasensors developed recently in literature, not only for food samples. Few more examples should be added based on the novel nanomaterials.

Response: Thanks for the reviewer’s advice. We have added several corresponding references (shown as following) regarding the recently developed electrochemical aptasensors in the Introduction section in the revised manuscript, especially some literatures based on several nanomaterials-based electrochemical aptasensors, such as A polyethyleneimine reduced graphene oxide/gold nanocubes [18], Ag nanoparticles sensitized bismuth oxyiodide [19], and Gold Nanoparticle-Functionalized 3D Flower-like MoS2/TiO2 Heterostructure [20], have been reported for CAP detection [p.2, line 89-96].

  1. Lu M.; Cao C.; Wang F.; Liu G. A polyethyleneimine reduced graphene oxide/gold nanocubes based electrochemical aptasensor for chloramphenicol detection using single-stranded DNA-binding protein. Mater Design. 2021, 199, 109409. https://https://doi.org/10.1016/j.matdes.2020.109409.
  2. Dong J.; Chen F.; Xu L.; Yan P.; Qian J.; Chen Y.; Yang M.; Li H. Fabrication of sensitive photoelectrochemical aptasensor using Ag nanoparticles sensitized bismuth oxyiodide for determination of chloramphenicol. Microchem J. 2022, 107317. https://https://doi.org/10.1016/j.microc.2022.107317.
  3. Zhao C.; Jing T.; Dong M.; Pan D.; Guo J.; Tian J.; Wu M.; Naik N.; Huang M.; Guo Z. A Visible Light Driven Photoelectrochemical Chloramphenicol Aptasensor Based on a Gold Nanoparticle-Functionalized 3D Flower-like MoS2/TiO2 Langmuir. 2022, 38(7), 2276-2286. https://10.1021/acs.langmuir.1c02956.

  1. Chapter 2.3. - Electrochemical characterization of the aptasensor - Here and neither in Chapter 3 Material and methods aren't presented the specific parameters used for EIS (frequency range, amplitude, etc) and for DPV (working applied potential, etc).

Response: Thank the reviewer for the suggestion. The specific parameters used for EIS and DPV of the electrochemical characterization of the aptasensor have been described in detail in Section 3.6 Electrochemical measurements in our revised manuscript as following [p.13, line 435-445].

Electrochemical impedance spectroscopy (EIS) measurements were performed in 5.0 mM K3[Fe(CN)6]/K4[Fe(CN)6] solution containing 0.10 M KCl with a frequency range of 0.01 Hz to 100 kHz and an amplitude of 5.0 mV. Differential pulse voltammetry (DPV) measurements were recorded in 5.0 mM K3[Fe(CN)6]/K4[Fe(CN)6] solution containing 0.10 M KCl in the potential range of -0.2-0.6 V. Cyclic voltammetry (CV) measurements (potential range of -0.2-0.6 V, scan rate of 100 mV s-1) were obtained to test the performance of the prepared electrodes.

  1. Chapter 2.4 - Optimization of experimental conditions - The specific procedures for preparation of this aptasensor require long time, more than 6 hour, which is a big disadvantage, and since there are not presented studies regarding the stability and reproducibility of this developed aptasensor, raises doubts about the accuracy of the analytical parameters obtained. 

Response: Thank the reviewer for the suggestion. We agree that the preparation procedure of the aptasensor especially the incubation of the aptamer requires a relative long time in order to obtain a stable sensing interface. Compared with many reported electrochemical aptasensors (shown as following) in which even more preparation time needed. By comparison, the 6 hours we have optimized is reasonable and acceptable.

[1] Zhao, Y.; Xu, Y.; Zhang, M.; Xiang, J.; Deng, C.; and Wu, H. An electrochemical dual-signaling aptasensor for the ultrasensitive detection of insulin. Anal. Biochem. 2019, 573, 30-36. 10.1016/j.ab.2019.02.032.

[2] Babamiri, B.; Salimi, A.; and Hallaj, R. Switchable electrochemiluminescence aptasensor coupled with resonance energy transfer for selective attomolar detection of Hg2+ via CdTe@CdS/dendrimer probe and Au nanoparticle quencher. Biosens and Bioelectronics. 2018, 102, 328-335. 10.1016/j.bios.2017.11.034.

[3] Li, F.; Wang, X.; Sun, X.; and Guo, Y. Multiplex electrochemical aptasensor for detecting multiple antibiotics residues based on carbon fiber and mesoporous carbon-gold nanoparticles. Sens. Actuators B Chem. 2018, 265, 217-226. 10.1016/j.snb.2018.03.042.

What’s more, we have investigated the reproducibility and stability of the constructed electrochemical aptasensor as shown in Fig. 5C and 5D. The results demonstrate that the proposed electrochemical aptasensor possesses good reproducibility and stability, indicating that the relative long preparation time did not influence the application of the aptasensor. The results have been added in Section 2.6 and shown as following.

  1. Chapter 2.5. - Should be completed with stability and reproducibility of the developed aptasensor, the storage life of it, etc.

Response: According to the reviewer’s suggestion, we have added the reproducibility and stability of the developed electrochemical aptasensor in Section 2.6. Selectivity, reproducibility and stability investigation (shown as following) as can been seen in red in the revised manuscript [p.9, line 331-343].

Fig. 5 (A) The inhibition ratio of the aptasensor in presence of 1 nM CAP or 50 nM interferences (MAG, MID, INH and DES), (B) DPV response of the aptasensor in presence of 1 nM CAP or 50 nM interferences (MAG, MID, INH and DES), (C) Reproducibility of the aptasensor, (D) Stability of the aptasensor.

   The reproducibility of the biosensor was evaluated by using five different GCEs to detect 1 nM CAP with the identical manufacturing processes (three measurements for each electrode). The relative standard deviation is less than 3.59%, indicating the designed sensing strategy has good reproducibility (seen in Fig. 5C).

For long-term stability investigation, a batch of electrodes were treated with the same fabrication procedure to obtain the sensing interface and stored in the refrigerator. The 1 nM CAP sample was measured day by day and the results were shown in Fig. 5D. After 15 days, the signal value remained 95% of the initial value, demonstrating excellent stability of developed aptasensor.

  1. Chapter 2.6. Selectivity studies - Figure 5 should be completed also with DPV (currents plots) for specific species using the reported concentrations.

Response: Thank the reviewer for the suggestion. The DPV currents plots have been added as shown in Fig.5B in the revised manuscript [p.9, line 325-327].

Fig. 5B DPV response of the aptasensor in presence of 1 nM CAP or 50 nM interferences (MAG, MID, INH and DES)

  1. Chapter 2.7.  I recommend validation test with another technique for the real samples of honey and spiked honey (pM concentrations).

Response: Thank the reviewer for the suggestion. To further evaluate the accuracy of the aptasensor, the result was compared with that from high performance liquid chromatography (HPLC), a standard detection method for CAP detection. As shown in Table 2, the results from our designed method are in good agreement with HPLC method. As expected, the designed method can be applied to determine the CAP concentration in spiked honey samples.

Table 2 Recovery measurements of CAP in spiked honey samples using the constructed aptasensor and HPLC method (n = 5).

Samples

Added

(pM)

Founded

(pM)

Recovery

(%)

by HPLC

method

0

0

0.02

-

0.018

1

50

53

106.0

54.85

2

500

483

96.6

512.70

3

1000

983

98.3

1024.01

4

5000

4788

95.7

4906.83

  1. Chapter 3 - Should be completed with correct information about electrochemical system (reference and counter electrodes used), specific parameters used for EIS, DPV and CV (potential range, scan rate, etc).

Response: Thank the reviewer for the suggestion. The specific parameters used EIS, DPV and CV of the electrochemical characterization of the aptasensor have been added in Section 3.6 Electrochemical measurements as shown in red in the revised manuscript [p.13, line 435-445].

Electrochemical impedance spectroscopy (EIS) measurements were performed in 5.0 mM K3[Fe(CN)6]/K4[Fe(CN)6] solution containing 0.10 M KCl with a frequency range of 0.01 Hz to 100 kHz and an amplitude of 5.0 mV. Differential pulse voltammetry (DPV) measurements were recorded in 5.0 mM K3[Fe(CN)6]/K4[Fe(CN)6] solution containing 0.10 M KCl in the potential range of -0.2-0.6 V. Cyclic voltammetry (CV) measurements (potential range of -0.2-0.6 V, scan rate of 100 mV s-1) were obtained to test the performance of the prepared electrodes.

  1. Chapter 4 - Should be completed in accordance with information about stability, reproducibility, response time and life time of the developed aptasensors, and which are the advantages of presented aptasensor compared with reported ones, in terms not only of limit of detection and linear range.

Response: Thanks for the valuable advice. We have added the advantages of the presented aptasensor such as the low detection limit, wide linear range and good reproducibility and stability in the conclusion section as show in red in the revised manuscript [p.9, line 331-343].

Reviewer 3 Report

The abbreviation   MXene must be briefly explained in abstract section.

Line 57-58. As recognition elements it is possible to use molecularly imprinted polymers. It is really suitable for small molecules such as drugs (chloramphenicol, for example).

Line 68.  For better understanding of the privilege of new group of metal carbides, nitrides, and carbonitrides known as MXene it is necessary to add explanation of the unique properties of these classes of modifies.

Experimental section. It is not clear how authors used as prepared sensor. Is there successive addition of growing concentration of chloramphenicol to the same modified electrode or for each concentration it was the new one?

Line 70. What means Ti3C2Tx? On Fig.1 authors investigate Ti3C2 MXene chemical structures and Ti3C2 MXene with incorporated AuNP.

The full text should be carefully checked for improvement of English language.

Author Response

Comments of reviewer #3 as well as our revisions and replies are as follows:

  1. The abbreviation MXene must be briefly explained in abstract section.

Response: Thanks for the reviewer’s kind suggestion. MXene has been explained in abstract section [p.1, line 22].

  1. Line 57-58. As recognition elements it is possible to use molecularly imprinted polymers. It is really suitable for small molecules such as drugs (chloramphenicol, for example).

Response: As an identifying element, molecularly imprinted polymers are indeed an effective strategy. In the present work, aptamers were used as recognition elements on MXene-AuNPs modified electrodes for the detection of chloramphenicol since thiol modified aptamer oligonucleotide is easy to immobilize on the AuNPs through S-Au bond. Upon specifically binding with CAP with high affinity, the CAP-Apt complexes produced low conductivity on the aptasensor surface, leading to a decreased electrochemical signal, which can be applied for sensitive detection of chloramphenicol in food. The aptasensor presented exhibits high sensitivity and selectivity with an estimated LOD of 0.03 pM and good reproducibility and stability.

  1. Line 68. For better understanding of the privilege of new group of metal carbides, nitrides, and carbonitrides known as MXene it is necessary to add explanation of the unique properties of these classes of modifies.

Response: Thanks for the valuable advice. The following sentence has been added to explain the privilege of MXene as shown in red in the Introduction section of revised manuscript [p.3, line 98-102].

Recently, two-dimensional (2D) MXenes have received great interest for their large group of carbides, nitrides, and carbonitrides and their unique properties such as high electrical conductivity, large specific surface areas and superior electrochemical properties.

  1. Experimental section. It is not clear how authors used as prepared sensor. Is there successive addition of growing concentration of chloramphenicol to the same modified electrode or for each concentration it was the new one?

Response: Thanks for the reviewer’s kind suggestion. As shown in the Experimental section and Results and Discussion section, one can see that the preparation procedure of the aptasensor is easy and good reproducibility and stability are obtained, so we used a new modified electrode for each concentration of sample.

  1. Line 70. What means Ti3C2Tx? On Fig.1 authors investigate Ti3C2 MXene chemical structures and Ti3C2 MXene with incorporated AuNP.

Response: Ti3C2Tx is the most widely studied type of MXene where Tx represents the functional groups on the surface of MXene nanomaterials such as –(OH)x and –Fx. We have revised the expression of MXene in Figure 1 to maintain consistency with that in Introduction Section.

  1. The full text should be carefully checked for improvement of English language.

Response: Thanks for the reviewer’s kind suggestion. The manuscript has been carefully checked by a native speaker to improve the English language.

Round 2

Reviewer 1 Report

The authors have addressed all my comments in detail and the manuscript can be accepted in its current form.

Reviewer 2 Report

The authors addressed all required aspects, thus the present revised form of the manuscript may be consider for publication.